# Post-Coronary Artery Bypass Grafting Outcomes of Patients with/without Type-2 Diabetes Mellitus and Chronic Kidney Disease Treated with SGLT2 Inhibitor Dapagliflozin: A Single-Center Experience Analysis

**DOI:** 10.3390/diagnostics14010016

**Published:** 2023-12-21

**Authors:** Razan Al Namat, Letiția Doina Duceac, Liliana Chelaru, Marius Gabriel Dabija, Cristian Guțu, Constantin Marcu, Maria Valentina Popa, Florina Popa, Elena Roxana Bogdan Goroftei, Elena Țarcă

**Affiliations:** 1Faculty of Medicine, “Grigore T. Popa” University of Medicine and Pharmacy Iași, 700115 Iași, Romania; al-namat.razan@umfiasi.ro (R.A.N.); liliana.chelaru@umfiasi.ro (L.C.); elena.goroftei@yahoo.com (E.R.B.G.); 2Faculty of Medicine and Pharmacy, “Dunărea de Jos” University, 800008 Galați, Romania; cristian.gutu@gmail.com (C.G.); constantin.marcu@yahoo.com (C.M.); valentina.popa@gmail.com (M.V.P.); florinapopa11@yahoo.com (F.P.); 3Department of Surgery II—Pediatric Surgery, “Grigore T. Popa” University of Medicine and Pharmacy, 700115 Iași, Romania; tarca.elena@umfiasi.ro

**Keywords:** CABG, heart failure, SGLT2 inhibitors, type-2 diabetes mellitus, H-FABP

## Abstract

Introduction: Increasingly, SGLT2 inhibitors save patients with heart failure and comorbidities such as type-2 diabetes mellitus (T2DM) and chronic kidney disease (CKD); the inhibition of sodium-glucose cotransporter 2 (SGLT2) was first studied in patients with diabetes as a solution to lower glucose levels by preventing glucose reabsorption and facilitating its elimination; in the process, researchers took notice of how SGLT2 inhibitors also seemed to have beneficial cardiovascular effects in patients with both diabetes and cardiovascular disease. Aim: Our single-center prospective study assesses outcomes of post-coronary artery bypass grafting (CABG) rehabilitation and SLGT2 inhibition in CABG patients with/without T2DM and with/without CKD. Materials and Methods: One hundred twenty consecutive patients undergoing CABG were included in the analysis. Patients were divided into four subgroups: diabetes patients with chronic kidney disease (T2DM + CKD), diabetes patients without chronic kidney disease (T2DM−CKD), prediabetes patients with chronic kidney disease (PreD+CKD), and prediabetes patients without chronic kidney disease (PreD−CKD). Echocardiographic and laboratory investigations post-surgery (phase I) and 6 months later (phase II) included markers for cardiac ischemia, glycemic status, and renal function, and metabolic equivalents were investigated. Results: One hundred twenty patients participated, mostly men, overweight/obese, hypertensive, smokers; 65 had T2DM (18 with CKD), and 55 were prediabetic (17 with CKD). The mean ejection fraction increased by 8.43% overall but significantly more in the prediabetes group compared to the T2DM group (10.14% vs. 6.98%, *p* < 0.05). Overall, mean heart-type fatty-acid-binding protein (H-FABP) levels returned to normal levels, dropping from 68.40 ng/mL to 4.82 ng/mL (*p* = 0.000), and troponin data were more nuanced relative to an overall, strongly significant decrease of 44,458 ng/L (*p* = 0.000). Troponin levels in patients with CKD dropped more, both in the presence of T2DM (by 82,500 ng/L, *p* = 0.000) and in patients without T2DM (by 73,294 ng/L, *p* = 0.047). As expected, the overall glycated hemoglobin (HbA1c) levels improved significantly in those with prediabetes (from 6.54% to 5.55%, *p* = 0.000); on the other hand, the mean HbA1c changed from 7.06% to 6.06% (*p* = 0.000) in T2DM, and the presence or absence of CKD did not seem to make any difference: T2DM+CKD 7.01–6.08% (*p* = 0.000), T2DM−CKD 7.08–6.04% (*p* = 0.000), PreD+CKD 5.66–4.98% (*p* = 0.014), and PreD−CKD 6.03–4.94% (*p* = 0.00). Compared to an overall gain of 11.51, the GFRs of patients with CKD improved by 18.93 (68.15–87.07%, *p* = 0.000) in the presence of established diabetes and 14.89 (64.75–79.64%, *p* = 0.000) in the prediabetes group. Conclusions: Regarding the patients’ cardiac statuses, the results from our single-center analysis revealed a significant decrease in ischemic risk (H-FABP and hs-cTnI levels) with improvements in mean ejection fraction, glycemic status, and renal function in patients post-CABG with/without T2DM, with/without CKD, and with SGLT2 inhibitor dapagliflozin treatment while undergoing cardiac rehabilitation.

## 1. Introduction

Epidemiological studies have shown that coronary artery disease is not only more common but also more advanced in patients with diabetes. In such cases, the benefits of the more invasive CABG-type surgery outweigh those of less invasive percutaneous interventions, which can be canceled by intrastent restenosis [1,2].

The international, multi-center, double-blind, randomized trial EMPA-REG OUTCOME, conducted between 2010 and 2015 on more than 7000 patients, demonstrated that the SGLT2 inhibitor empagliflozin can reduce the risk of complications and death substantially compared to a placebo. The US Food and Drug Administration consequently approved empagliflozin for T2DM patients with established cardiovascular disease, and authoritative professional associations of diabetologists began recommending SGLT2 inhibition in their therapeutic guidelines for practitioners [3,4,5].

Another example of a large, multi-center, randomized trial assessing SGLT2 inhibition is DECLARE-TIMI58, where 17,160 patients with type-2 diabetes were given either dapagliflozin or a placebo. This trial was conducted from 2013 to 2018, and the patients were monitored for a median of 4.2 years. Overall, lower rates of cardiovascular death and hospitalization for heart failure were reported in the experimental versus the placebo groups [6,7,8]. Expanding on the initial diabetes-related studies and applications of SGLT2 inhibitors, further research has shown how such drugs can facilitate positive outcomes in cardiovascular patients independent of diabetes status. Notably, the DAPA-HF placebo-based randomized trial conducted between 2017 and 2019 enrolled 4744 patients with chronic heart failure with reduced ejection fractions. Of them, 55% entered the study without established diabetes, but two-thirds turned out to have glycated hemoglobin levels between 5.7 and 6.4%, indicative of prediabetes. During the 18 months of administration and monitoring, significantly fewer incidents of worsening heart failure or even death were recorded in the three groups of diabetes, prediabetes, and non-diabetes patients treated with the SGLT2 inhibitor dapagliflozin compared to matching groups of patients receiving a placebo [9,10,11]. More noteworthy effects of SGLT2 inhibitors involve kidney function. This is relevant because renal dysfunction manifested as a low/decreasing glomerular filtration rate is known to exacerbate cardiovascular risk, especially in combination with type-2 diabetes [12,13]. Since SGLT2 inhibitors facilitate the excretion of glucose via urine, it made sense initially that impaired kidney function would undermine the intended glucose management targets. This is why the US FDA approved the use of SGLT2 inhibitors for patients with estimated GFRs of at least 45 mL/min/1.73 m^2^. In the meantime, however, it has become apparent that a low GFR does not inhibit other favorable outcomes.

In fact, preserving renal function and slowing down the progression of kidney disease are in themselves valuable effects of SGLT2 inhibitors for which there is robust evidence, according to a recent meta-analysis and a state-of-the-art review [14,15]. In the international CREDENCE trial on 4401 patients with a history of type-2 diabetes and kidney disease, the administration of the SGLT2 inhibitor canagliflozin resulted in fewer instances of both kidney failure and cardiovascular events compared to the placebo group during a median follow-up period of 2.62 years [16]. Moreover, a secondary analysis of the aforementioned DECLARE-TIMI58 trial data focusing on the participants’ renal function revealed that the cardiovascular protection of dapagliflozin was actually the highest among patients who started off with reduced GFRs [17]. Heart-type fatty-acid-binding protein (H-FABP), as well as other non-invasive biomarkers, like highly sensitive troponin T (hsTnT), were found to be an important indicator of myocardial injury and major adverse cardiovascular events (MACEs) with high sensitivity and specificity [18].

The positive results accumulating in the literature and the formal approval of dapagliflozin in Romania in 2007 influenced our clinical decision to include this SGLT2 inhibitor in the post-CABG treatment of patients with complex cardiovascular, metabolic, and renal pathologies. This study aims to assess the effects of six months of standard cardiac rehabilitation provision and concurrent treatment with dapagliflozin in post-CABG patients with type-2 diabetes mellitus or elevated glycated hemoglobin indicative of prediabetes in the presence or absence of chronic kidney disease and the effect of the SGLT2 inhibitor on the ischemic risk (H-FABP and troponin levels) in CABG patients undergoing cardiac rehabilitation.

## 2. Materials and Methods

### 2.1. Overview of Study Coordinates and Phases

The study enrolled consecutive consenting patients who underwent on-pump CABG surgery at the “Prof. Dr. George I.M. Georgescu” Institute of Cardiovascular Diseases in the north-east Romanian city of Iași during 2018–2020. After their interventions and initial recovery, the patients were transferred to the partner hospital from Iași specialized in clinical rehabilitation. Formal approval for the study was obtained from the Ethics Committee of the “Grigore T. Popa” University of Medicine and Pharmacy Iași, and the general provisions of the Declaration of Helsinki regarding medical research on human subjects were observed. The study was conducted prospectively over a period of 6 months. In addition, medical history data were retrieved from the patients’ records. In the first hours following the intervention, samples were collected for routine tests and for the purpose of measuring H-FABP levels. We call this phase I of the study. After completion of a six-month cardiac rehabilitation program, a similar battery of tests provided the follow-up data, and we refer to this as phase II.

### 2.2. Inclusion and Exclusion Criteria

Patient enrollment was based on written informed consent; age of 40 years or older; and the clinical indication for CABG surgery established upon admission considering the patient’s symptoms, echocardiographic evidence of heart failure, angiographic evidence of artery disease, and SYNTAX Score II. A previously established diagnosis of type-2 diabetes mellitus, glycated hemoglobin levels indicative of prediabetes upon admission, or both were also of interest in recruiting participants. The exclusion criteria were as follows: class IV NYHA heart failure, lack of consent or impaired ability due to psychiatric conditions or other reasons, physical disability limiting engagement in the physical rehabilitation program, poor compliance or dropping out of the rehabilitation program for any reason, advanced chronic kidney disease (stage 5 dialysis patients) or neoplasms.

### 2.3. Treatment and Rehabilitation

Dapagliflozin was administered at daily doses of 10 mg. Otherwise, the medication plans were individualized based on the patients’ progress and other comorbidities (T2DM and related treatment). The cardiac post-CABG rehabilitation proceeded in a series of stages. In phase I, a few days after the surgery, the patients were taught to perform breathing exercises and low-intensity movements. Then, in phase II, they were transferred to the partner hospital specializing in clinical recovery, where they participated in regular kinesiotherapy sessions.

### 2.4. Patient Data Collected and Definitions

The collected data were organized into sets and series as can be seen in the Results section for the studied variables:General patient characteristics and past medical history;Clinical, paraclinical, echocardiographic, and laboratory findings in phase I and then again in phase II.

Also, widely accepted definitions and thresholds were used to assess the key characteristics and outcomes in the study, such as the following:Heart failure upon admission based on left ventricular ejection fraction values: heart failure with reduced ejection fraction (severely reduced if EF ≤ 30%, moderately reduced if EF is between 31% and 40%, mildly reduced if EF is between 41% and 49%), and heart failure with preserved ejection fraction if EF ≥ 50% [18];Type-2 diabetes mellitus or prediabetes as had been diagnosed prior to the study (part of the patient’s medical history data), or prediabetes diagnosed incidentally based on glycated hemoglobin levels between 5.7% and 6.4% at the time of the surgery.

The myocardial ischemia biomarker H-FABP was sampled in the first 6 h after CABG and again after 6 months using Randox kits as per the manufacturer’s instructions and the latex-enhanced immunoturbidimetric assay method.

### 2.5. Organization of Database and Statistical Analysis

The patient data were first anonymized and collected in Microsoft Office Excel version 2010 and then processed in IBM SPSS Statistics for Windows, version 20 (IBM Corp., Armonk, NY, USA). The data series was organized into the following two main study groups and four subgroups based on the presence or absence of the two studied comorbidities:Patients with established type-2 diabetes mellitus (T2DM):
With chronic kidney disease as well (T2DM+CKD);Without chronic kidney disease (T2DM−CKD).
Patients with established or incidental prediabetes (PreD):
With chronic kidney disease as well (PreD+CKD);Without chronic kidney disease (PreD−CKD).


## 3. Results

### 3.1. General Patient Characteristics and Relevant Past Medical History

One hundred twenty patients met the inclusion criteria and were enrolled in the study. They were mostly men (75.83%) and mostly urban residents (88.33%). The patients’ mean age was 65.93 years; the youngest was 41 years old, and the oldest was 85 years old (median 67 years).

All but 9 patients had hypertension (92.5%), and 87 had a known history of heart failure (72.5%). Over a quarter of patients had peripheral artery disease (32 cases), and six had also suffered a stroke. In terms of general cardiovascular risk factors, the patients’ overall body mass indices averaged just above the obesity threshold (30.12 kg/m^2^), and most of the patients were smokers (60.83%).

Regarding the comorbidities of particular interest in this study, 65 patients had a known history of type-2 diabetes mellitus (51.17%), and 35 had established chronic kidney disease (29.17%). Eighteen patients had both conditions (15%). In what follows, the results are analyzed, presented, and discussed relative to the subgroups of patients with or without T2DM and CKD—see Table 1.

### 3.2. Echocardiographic and Laboratory Findings

The echocardiographic and laboratory findings post-CABG (phase I) and after six months of cardiac rehabilitation and treatment (phase II) are summarized in Table 2 and Table 3.

The skewness of all data series was inspected before the appropriate statistical significance tests were chosen. For example, ejection fraction values were only moderately skewed (0.27 in phase I and 0.52 in phase II), indicating normal distribution. By contrast, fasting glucose levels were highly skewed (2.68 in phase I and 2.08 in phase II) and, as such, non-normally distributed—see Figure 1 and Figure 2.

Regarding the patients’ cardiac status, the results from Table 2 indicate substantial improvements in the six months of post-CABG rehabilitation and treatment. The mean ejection fraction increased by 8.43% overall; it changed from 43.5% to 51.9% (*p* = 0.000), but significantly more in the prediabetes group compared to the T2DM group (10.14% vs. 6.98%, *p* < 0.05). However, the 6.06% ejection fraction gains in prediabetes patients with CKD, specifically, did not differ significantly from the 7.17% ejection fraction gains of diabetes patients with CKD, nor from the 6.92% increase in the T2DM subgroup without CKD. The ejection fractions of patients without either comorbidity increased by 11.98%, which was significantly more than in all other subgroups.

The results for the two biomarkers used to assess cardiac ischemia—H-FABP and high-sensitivity cardiac troponin (hs-cTnI)—show significant amelioration from phase I to phase II, as well as noteworthy differences depending on the two studied comorbidities. Overall, mean H-FABP levels returned to normal levels, dropping from 68.40 ng/mL to 4.82 ng/mL (*p* = 0.000). Interestingly, the prediabetic patients, especially those with chronic kidney disease, started with the highest H-FABP levels (85.15 ng/mL) which then normalized at 5.52, resulting in the most substantial phase I-II difference (79.63 ng/mL). By contrast, T2DM patients both with and without CKD had much lower H-FABP levels post-surgery (53.09 ng/mL and 61.67 ng/mL, respectively), yet after six months they achieved similarly reduced levels (4.74 ng/mL and 4.83 ng/mL, respectively).

Troponin data are more nuanced. Relative to an overall, strongly significant decrease of 44,458 ng/L (*p* = 0.000), troponin levels in patients with CKD dropped more, both in the presence of T2DM (by 82,500 ng/L, *p* = 0.000) and otherwise (by 73,294 ng/L, *p* = 0.047). Also, troponin decreased only insignificantly in patients without comorbidity (only 3842 ng/L, *p* = 0.102), in strong contrast to all other subgroups (marked with superscript letter a).

These significant decreases in H-FABP and hs-cTnI levels over the course of the six-month rehabilitation and treatment post-CABG suggest a favorable evolution regarding the patients’ initial myocardial ischemia and necrosis. There may have been unreported episodes of aggravation, but none of the patients enrolled in this study returned complaining of angina or presenting signs of reinfarction. Also, no cases of reinfarction were excluded from the study.

Next, we review outcomes related to the patients’ comorbidities—glycemic control and kidney function (see Table 3). These are not fixed input quantities, but dynamic pathological processes known and meant to interact favorably with the kind of rehabilitation and SGLT2-supplemented treatment prescribed to cardiac patients.

As expected, overall glycated hemoglobin levels improved significantly from slightly over to slightly under the range indicative of prediabetes (from 6.54% to 5.55%, *p* = 0.000). In the T2DM group, mean HbA1c changed from 7.06% to 6.06% (*p* = 0.000), and the presence or absence of CKD did not seem to make a difference. For the patients without any history of established diabetes upon enrollment, mean HbA1c reached below 5% for both subgroups (with/without CKD).

Fasting plasma glucose levels, on the other hand, differed much more between groups. Overall, mean levels decreased significantly from 138.53 mg/dL to 120.63 mg/dL, which is still high (*p* = 0.000). In the T2DM group, although the phase I–II improvement was greater (by 22.99 mg/dL, *p* = 0.000), mean glycemic levels remained elevated (141.77 mg/dL). Also, the presence of chronic kidney disease created a noticeable discrepancy within this group by the end of the studied period. Namely, patients with both comorbidities had a mean glycemia of 157.78 mg/dL, while T2DM-only patients achieved a mean of 135.64 mg/dL. However, it should be noted that the condition for statistical significance was not met by the glycemic levels of patients with CKD (see corresponding *p* values > 0.05 in Table 3).

For kidney function, the mean estimated glomerular filtration rates calculated with the MDRD-GFR formula were similar after surgery in both diabetes and prediabetes groups. Interestingly, patients with both comorbidities, in other words, the unhealthiest patients enrolled in the study, started off with a better mean GFR (68.15) than any other subgroup. Also, when we look at just the prediabetes group, established CKD did not manifest as necessarily a poorer GFR at the beginning. By the end of the study period, however, it was the patients with chronic kidney disease whose GFRs increased the most. Compared to an overall gain of 11.51, the GFRs of patients with CKD improved by 18.93 in the presence of established diabetes and 14.89 in the prediabetes group (*p* = 0.000).

When looking at the creatinine levels specifically, we noticed statistically significant improvement all around, but less marked in patients without CKD. Relative to the overall decrease of 0.20 mg/dL (*p* = 0.000), patients with CKD seemed to have benefitted more, especially in the absence of established diabetes (a decrease of 0.42 mg/dL from 1.53 mg/dL to 1.10 mg/dL, *p* = 0.042).

For urea, the most noticeable differences were between patients with and without diabetes. In phase II, the urea levels of prediabetic patients decreased three times as much as those of T2DM patients (by 6.67 mg/dL at *p* = 0.000 versus a modest and insignificant decrease of 1.68 mg/dL at *p* = 0.403). However, relative to age, for 16 of the 33 patients aged 60 or younger (48.48%), phase II urea levels were still higher than the recommended target for their age (<43 mg/dL). Similarly, 29 of the 87 patients older than 60 had better (33.33%) but still elevated levels of urea compared to the reference level for their age (<49 mg/dL). In other words, by the end of the studied period, the kidney function of patients without established diabetes appeared to have significantly improved, but urea levels were still sub-optimal in as many as 37.5% of cases.

Inflammatory status and lipid profile are the other two sets of parameters described in Table 3 for the purposes of this study, to see if established T2DM and/or CKD nuanced these types of outcomes. Overall, systemic inflammation decreased significantly during the six-month cardiac rehabilitation and treatment period. Initially elevated fibrinogen (636.96 mg/dL) approached normal levels in phase II (441.62 mg/dL). The most substantial improvement was recorded in prediabetic patients with CKD (a decrease of 231.73 mg/dL, *p* = 0.000), while the smallest benefit, yet still statistically significant, was obtained by the patients with both studied comorbidities (a decrease of only 92.39 mg/dL, *p* = 0.000). As for the levels of C-reactive protein, initial values ranging between 3 and 4 mg/L signaled the presence of inflammation in the body post-CABG. They then decreased to below 3 mg/L in all subgroups. The best results in terms of CRP reduction were in the T2DM group (from 3.64 to 1.07 mg/L, *p* = 0.000), and the most modest were in the case of the patients starting off without either diabetes or chronic kidney disease (from 3.96 to 2.72, *p* = 0.151).

Regarding the patients’ lipid profile, we notice in Table 3 that patients starting as the least unhealthy were also those whose cholesterol levels improved the least compared to patients with one or both comorbidities. For total cholesterol, phase I levels were not very concerning, and we see them improving the most in the diabetes group, especially the subgroup with both T2DM and CKD. Also, LDL-cholesterol levels improved significantly in all groups and especially in the T2DM group (from 138.83 to 115 mg/dL, *p* = 0.000), but it is worth noting that LDL-cholesterol still remained outside the recommended target of <100 mg/dL. At the same time, the so-called “good” HDL-cholesterol increased significantly in all groups and especially in patients with both comorbidities, but not enough to reach desirable levels.

The levels of triglycerides also improved, but the results summarized in Table 3 reveal statistically non-significant reductions in the presence of chronic kidney disease. The patients whose triglycerides dropped the most were the patients with diabetes only (from 160.32 md/dL to 139.01 mg/dL, *p* = 0.006). These were also the patients who started off with above-normal levels.

## 4. Discussion

This non-randomized study reports on the outcomes of 120 CABG patients who underwent a six-month cardiac rehabilitation program including treatment with the SGLT2 inhibitor dapagliflozin. All patients undergoing CABG who met the inclusion criteria were included in our study. The patient data were organized based on the presence or absence of established type-2 diabetes mellitus (T2DM) and chronic kidney disease (CKD): general characteristics, medical history, and echocardiographic and laboratory findings after surgery (phase I) and again after six months (phase II). The analysis focused on statistically significant differences between phase I and phase II, as well as patterns of significant associations with the two studied comorbidities and other phase I characteristics.

The cases with type-2 diabetes mellitus (T2DM) represent about one-third of the cases affected by cardiovascular conditions [19]. Coronary heart disease (CHD) is the most common type of cardiovascular disease (CVD) and the primary cause of death in T2DM cases [19,20]. Indeed, insulin resistance and hyperglycemia could lead to endothelial dysfunction and vascular complications via over-inflammation, causing a worse prognosis in T2DM than in non-T2DM cases [21]. In addition, T2DM vs. non-T2DM cases have an advanced rate of multivessel coronary stenosis, which causes plaque rupture, acute intracoronary thrombosis, and adverse clinical events [22]. In this environment, coronary artery bypass grafting (CABG) is a recommended revascularization strategy to ameliorate clinical issues in T2DM cases with multivessel coronary stenosis [1,23]. This over-inflammatory response could lead to a worse prognosis in CABG-treated cases, particularly in those with T2DM [1,24]. Indeed, the altered glucose homeostasis and insulin resistance could beget over-inflammation, linked to an increased expression of sodium-glucose transporter 2 receptors, and to worse prognosis post-CABG. In this environment, in T2DM cases treated with CABG, the sodium-glucose transporter 2 inhibitor (SGLT2i) “empagliflozin” leads to a profound reduction in cardiovascular mortality, hospitalization for heart failure, and incident or worsening nephropathy [25]. Still, an SGLT2i could be used for the secondary forestallment of cardiovascular events after CABG in individuals with T2DM. On the other hand, little is known about the beneficial effects of SGLT2i treatment on glucose homeostasis, over-inflammation, and clinical issues in T2DM patients treated with CABG [26].

According to the echocardiographic results, the ejection fraction increased by 8.43% overall and significantly more in the prediabetes group (10.14%), especially in the absence of CKD (11.98%). Also, the fact that comorbidities such as T2DM and CKD weigh down on such improvements is no surprise [27]. The cardiac ischemia marker H-FABP dropped within the normal range during the studied period (from 68.40 ng/mL to 4.82 ng/mL). Interestingly, the decrease in H-FABP from immediately after CABG to six months later correlated negatively and significantly with the number of grafts. The relationship between the number of CABG grafts and the evolution of myocardial ischemia early on has been reported in the literature [28], but this result seems to suggest that the association may continue for longer than expected, well into the months following the intervention [29,30,31].

Next, regarding renal function, post-CABG estimated glomerular filtration rates were similar in the T2DM and prediabetes groups, and all groups improved significantly, without any apparent deterrence by initial myocardial ischemia manifested as higher H-FABP. Within subgroups, patients with both diabetes and chronic kidney disease started with better GFRs, and interestingly, six months later, it was these initially unhealthiest patients who improved the most (by 18.93 compared to the overall mean gain of 11.51).

Systemic inflammation decreased significantly during the studied period. It is interesting to note that higher initial levels of inflammation markers resulted in their significantly inferior reduction six months later (based on strong-intensity negative correlation coefficients at *p* = 0.000). The most substantial fibrinogen reduction occurred in prediabetes patients with CKD, while the least important was the subgroup with both T2DM and CKD (also in the presence of peripheral artery disease). Also, there was a significant, even though weak, correlation between initial fibrinogen levels and the extent of H-FABP reduction, which points to the relationship between myocardial ischemia and systemic inflammation [32].

Last but not least, from the multivariate analysis of aggregate data identifying the unhealthiest from the least unhealthy patients, we found that certain outcomes were impacted significantly by this unfavorable profile. Namely, age, presence/absence of comorbidities (T2DM and CKD), and weight loss, taken together, explained a fair share of myocardial ischemia and inflammation improvements (41% of H-FABP, 31.5% of cardiac troponin, and 39.7% of fibrinogen reduction from phase I to phase II).

The decision to treat patients with an SGLT2 inhibitor had already been made on account of scientific evidence and clinical recommendations available at the time. This study does not seek to determine the impact of dapagliflozin, which has largely been established as beneficial for the pathologies considered here, as it would otherwise have needed a control group of patients not receiving this type of drug [33]. This study is the first study in the literature that highlights the impact of an SGLT2 inhibitor on CABG patients with/without T2DM and with/without CKD undergoing a rehabilitation program and compares the influence of an SGLT2 inhibitor on H-FABP levels. Some of the limitations are the small number of enrolled women and the fact that no provision was made from the start to enroll CABG recipients with adequate glycemic control (without either established diabetes or signs of prediabetes).

## 5. Conclusions

We know from the literature that in diabetes patients with a history of CABG surgery, the addition of an SGLT2 inhibitor to standard treatment reduces the risk of cardiovascular-related mortality and of further hospitalization for heart failure by half. Our single-center prospective study contributes a nuanced view of post-CABG rehabilitation outcomes in patients with/without T2DM and with/without CKD, pathologies for which an SGLT2 inhibitor has been proven beneficial during a cardiovascular rehabilitation program. Also, we demonstrate the importance of an SGLT2 inhibitor in patients undergoing CABG by showing that it reduces ischemic risk (by decreasing the H-FABP levels) in patients with/without T2DM and with/without CKD.

## Figures and Tables

**Figure 1 diagnostics-14-00016-f001:**
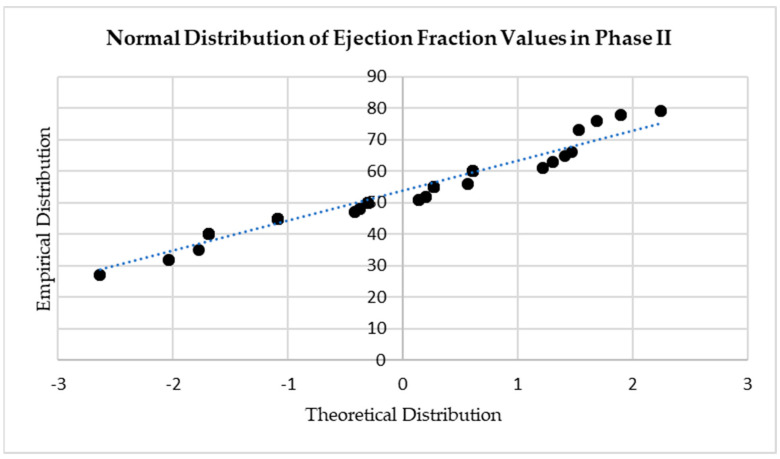
Q-Q plot of phase II ejection fraction values illustrating normal data distribution.

**Figure 2 diagnostics-14-00016-f002:**
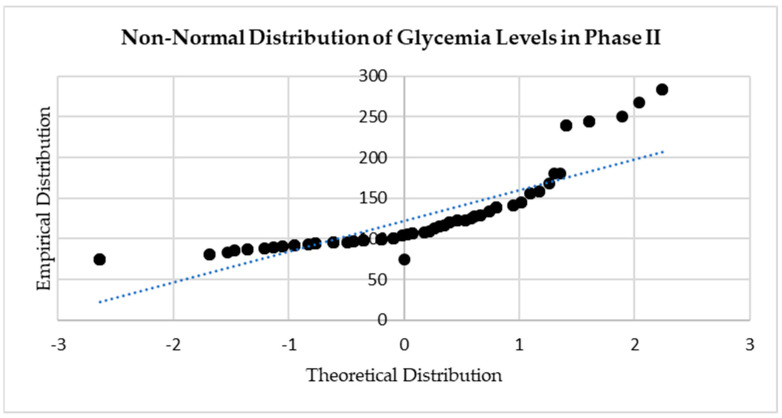
Q-Q plot of phase II glycemia values illustrating non-normal data distribution.

**Table 1 diagnostics-14-00016-t001:** General patient characteristics and medical history.

General Data	Overall (*N* = 120)	T2DM (*N* = 65)	T2DM+CKD (*N* = 18)	T2DM−CKD (*N* = 47)	PreD (*N* = 55)	PreD+CKD (*N* = 17)	PreD−CKD (*N* = 38)
Mean age (years)	65.93	67.25	70.22	66.11	64.36	65.47	63.87
Median age (years)	67	67	70	65	67	66	67.5
Age range (years)	41–85	58–85	58–80	46–85	55–77	55–75	41–77
Male sex	91 (75.83%)	53 (81.54%)	16 (88.89%)	37 (78.72%)	38 (69.09%)	15 (88.24%)	23 (60.53%)
Urban	106 (88.33%)	55 (84.62%)	17 (94.44%)	38 (80.85%)	51 (92.73%)	16 (94.12%)	35 (92.11%)
Smokers	73 (60.83%)	44 (67.69%)	12 (66.67%)	32 (68.09%)	29 (52.73%)	11 (64.71%)	18 (47.37%)
BMI (kg/m^2^)	30.12	30	30.78	29.70	28.73	30.12	28.11
**Hypertension**	**111 (92.5%)**	**65 (100%)**	**18 (100%)**	**47 (100%)**	**46 (83.64%)**	**15 (88.24%)**	**31 (81.58%)**
No history	9 (7.5%)	-	-	-	9 (16.36%)	2 (11.76%)	7 (18.42%)
Stage 1	7 (5.83%)	2 (3.08%)	-	2 (4.26%)	5 (9.09%)	2 (11.76%)	3 (7.89%)
Stage 2	22 (18.33%)	14 (21.54%)	2 (11.11%)	12 (25.53%)	8 (14.55%)	2 (11.76%)	6 (15.79%)
Stage 3	82 (68.33%)	49 (75.38%)	16 (88.89%)	33 (70.21%)	33 (60.00%)	11 (64.71%)	22 (57.89%)
**Heart failure**	**87 (72.5%)**	**49 (75.38%)**	**12 (66.67%)**	**37 (78.72%)**	**38 (69.09%)**	**15 (88.24%)**	**23 (60.53%)**
No history	33 (27.5%)	16 (24.62%)	6 (33.33%)	10 (21.28%)	17 (30.91%)	2 (11.76%)	15 (39.47%)
Class I	6 (5%)	2 (3.08%)	1 (5.56%)	1 (2.13%)	4 (7.27%)	1 (5.88%)	3 (7.89%)
Class II	33 (27.5%)	20 (30.77%)	3 (16.67%)	17 (36.17%)	13 (23.64%)	6 (35.29%)	7 (18.42%)
Class III	48 (40%)	27 (41.54%)	8 (44.44%)	19 (40.43%)	21 (38.18%)	8 (47.06%)	13 (34.21%)
**Peripheral artery disease**	**32 (26.67%)**	**24 (36.92%)**	**10 (55.56%)**	**14 (29.79%)**	**8 (14.55%)**	**4 (23.53%)**	**4 (10.53%)**
**Chronic kidney disease**	**35 (29.17%)**	**18 (27.69%)**	**18 (100%)**	**-**	**17 (30.91%)**	**17 (100%)**	**-**
Stage 1	28 (23.33%)	15 (23.08%)	15 (83.33%)	-	13 (23.64%)	13 (76.47%)	-
Stage 2	3 (2.5%)	3 (4.62%)	3 (16.67%)	-	-	-	-
Stage 3	4 (3.33%)	-	-	-	4 (7.27%)	4 (23.53%)	-
Stage 4	-	-	-	-	-	-	-
**Stroke**	**6 (5%)**	**-**	**-**	**2 (4.26%)**	**4 (7.27%)**	**-**	**4 (10.53%)**

T2DM—main study group of patients with established type-2 diabetes mellitus; T2DM+CKD—subgroup of diabetes patients with chronic kidney disease; T2DM−CKD—subgroup of diabetes patients without chronic kidney disease; PreD—main study group of patients with glycated hemoglobin levels indicative of prediabetes; PreD+CKD—subgroup of prediabetes patients with chronic kidney disease; PreD−CKD—subgroup of prediabetes patients without chronic kidney disease.

**Table 2 diagnostics-14-00016-t002:** Echocardiographic and laboratory findings related to cardiac status from phase I to II.

Studied Variable	Overall (*N* = 120)	*p* Value	T2DM (*N* = 65)	*p* Value	T2DM+ CKD (*N* = 18)	*p* Value	T2DM−CKD (*N* = 47)	*p* Value	PreD (*N* = 55)	*p* Value	PreD+ CKD (*N* = 17)	*p* Value	PreD− CKD (*N* = 38)	*p* Value
Cardiac status														
EF I	43.5	0.000	44.77	0.000	44.22	0.000	44.98	0.000	41.93	0.000	47.59	0.000	39.39	0.000
EF II	51.9		51.75		51.39		51.89		52.07		53.65		51.37	
Δ EF	↑ 8.43		↑ 6.98		↑ 7.17		↑ 6.92 ^(c)^		↑ 10.14 ^(b)^		↑ 6.06 ^(c)(c)^		↑ 11.98 ^(c)(b)(b)^	
FS I	26.08	0.074	26.22	0.722	25.44	0.143	26.51	0.967	25.93	0.025	28.29	0.475	24.87	0.020
FS II	27.57		26.62		26.72		26.57		28.69		30.24		28.00	
Δ FS	↑ 1.48		↑ 0.40		↑ 1.28		↑ 0.06 ^(c)^		↑ 2.76 ^(c)^		↑ 1.94 ^(c)(b)^		↑ 3.13 ^(c)(b)(c)^	
LVMi I	143.31	0.000	150.29	0.000	167.94	0.000	143.53	0.000	135.05	0.000	119.94	0.000	141.82	0.000
LVMi II	119.98		128.74		142.83		123.34		109.64		97.76		114.95	
Δ LVMi	↓ 23.33		↓ 21.55		↓ 25.11		↓ 20.19 ^(c)^		↓ 25.414 ^(c)^		↓ 22.18 ^(c)(c)^		↓ 26.87 ^(c)(b)(c)^	
Myocardial cytolytic enzymes														
H-FABP I	68.40	0.000	59.29	0.000	53.09	0.000	61.67	0.000	79.16	0.000	85.15	0.000	76.48	0.000
H-FABP II	4.82		4.81		4.74		4.83		4.83		5.52		4.53	
Δ H-FABP	↓ 63.58		↓ 54.48		↓ 48.35		↓ 56.83 ^(b)^		↓ 74.33 ^(b)^		↓ 79.63 ^(b)(b)^		↓ 71.95 ^(b)(b)(c)^	
hs-cTn I	361,375	0.000	357,076.9	0.003	321,722.2	0.000	370,617	0.173	366,454.5	0.012	378,882.4	0.047	360,894.74	0.102
hs-cTn II	316,916.7		316,630.8		239,222.2		346,276.6		317,254.5		305,588.2		322,473.68	
Δ Troponin	↓ 44,458		↓ 40,446		↓ 82,500		↓ 24,340 ^(b)^		↓ 49,200 ^(c)^		↓ 73,294 ^(c)(a)^		↓ 3842 ^(a)(c)(a)^	

Abbreviations and measuring units: EF—ejection fraction (%); FS—fractional shortening (%); LVMi—left ventricular mass index (g/m^2^); H-FABP—heart-type fatty-acid-binding protein (ng/mL); hs-cTn—high-sensitivity cardiac troponin (ng/L). Symbols: Δ—difference from phase I to phase II; ↑ and ↓—increasing/decreasing levels. Superscript: ^(a)^ *p* < 0.001; ^(b)^ *p* < 0.05; ^(c)^ *p* > 0.05 for differences between the study (sub)groups.

**Table 3 diagnostics-14-00016-t003:** Other laboratory findings related to glycemic, renal, inflammation, and lipid status.

Studied Variable	Overall (*N* = 120)	*p* Value	T2DM (*N* = 65)	*p* Value	T2DM+ CKD (*N* = 18)	*p* Value	T2DM− CKD (*N* = 47)	*p* Value	PreD (*N* = 55)	*p* Value	PreD+ CKD (*N* = 17)	*p* Value	PreD− CKD (*N* = 38)	*p* Value
Diabetes biomarkers														
HbA1c I	6.54	0.000	7.06	0.000	7.011	0.000	7.08	0.000	5.92	0.000	5.66	0.014	6.03	0.000
HbA1c II	5.55		6.06		6.083		6.048		4.95		4.98		4.94	
Δ HbA1c	↓ 0.99		↓ 1.0		↓ 0.93		↓ 1.03 ^(c)^		↓ 0.97 ^(c)^		↓ 0.68 ^(c)(c)^		↓ 1.10 ^(c)(c)(b)^	
Glycemia I	138.53	0.000	164.75	0.000	165.89	0.089	164.32	0.000	107.55	0.000	104.12	0.616	109.08	0.000
Glycemia II	120.63		141.77		157.78		135.64		95.65		102.24		92.71	
Δ Glycemia	↓ 17.90		↓ 22.99		↓ 8.11		↓ 28.68 ^(c)^		↓ 11.89 ^(b)^		↓ 1.88 ^(a)(a)^		↓ 16.37 ^(b)(b)(a)^	
Kidney function														
MDRD-GFR I	65.50	0.000	65.20	0.000	68.15	0.000	64.07	0.001	65.85	0.000	64.75	0.000	66.35	0.003
MDRD-GFR II	77.01		75.54		87.07		71.12		78.76		79.64		78.36	
Δ MDRD-GFR	↑ 11.51		↑ 10.34		↑ 18.93		↑ 7.05 ^(b)^		↑ 12.91 ^(c)^		↑ 14.89 ^(c)(b)^		↑ 12.01 ^(c)(c)(c)^	
Urea I	45.42	0.000	42.91	0.191	51.72	0.403	39.53	0.327	48.38	0.000	45.41	0.000	49.71	0.001
Urea II	41.45		41.23		49.17		38.19		41.71		39.24		42.82	
Δ Urea	↓ 3.97		↓ 1.68		↓ 2.56		↓ 1.34 ^(c)^		↓ 6.67 ^(b)^		↓ 6.18 ^(a)(a)^		↓ 6.90 ^(b)(b)(c)^	
Creatinine I	1.28	0.000	1.26	0.000	1.24	0.001	1.28	0.001	1.30	0.002	1.53	0.042	1.20	0.012
Creatinine II	1.09		1.10		0.97		1.15		1.06		1.10		1.04	
Δ Creatinine	↓ 0.20		↓ 0.16		↓ 0.26		↓ 0.12 ^(c)^		↓ 0.24 ^(c)^		↓ 0.42 ^(c)(c)^		↓ 0.15 ^(c)(c)(c)^	
Inflammatory status														
Fibrinogen I	636.96	0.000	624.78	0.000	570.89	0.000	645.43	0.000	651.35	0.000	549.29	0.000	697.00	0.000
Fibrinogen II	441.62		460.23		478.50		453.23		419.62		368.94		442.29	
Δ Fibrinogen	↓ 195.34		↓ 164.55		↓ 92.39		↓ 192.19 ^(b)^		↓ 231.73 ^(b)^		↓ 180.35 ^(a)(c)^		↓ 254.7 ^(a)(b)(b)^	
CRP I	3.74	0.000	3.64	0.000	3.37	0.000	3.75	0.000	3.84	0.011	3.57	0.000	3.96	0.151
CRP II	1.62		1.07		1.19		1.03		2.26		1.22		2.72	
Δ CRP	↓ 2.12		↓ 2.57		↓ 2.18		↓ 2.72 ^(c)^		↓ 1.58 ^(c)^		↓ 2.35 ^(c)(c)^		↓ 1.24 ^(b)(b)(b)^	
Lipid profile														
Total chol I	181.54	0.000	185.89	0.000	178.44	0.001	188.74	0.000	176.40	0.445	186.59	0.003	171.84	0.896
Total chol II	168.10		165.34		152.06		170.43		171.36		167.65		173.03	
Δ Total chol	↓ 13.42		↓ 20.55		↓ 26.39		↓ 18.32 ^(c)^		↓ 5.04 ^(b)^		↓ 18.94 ^(b)(c)^		↑ 1.18 ^(a)(a)(a)^	
LDL-chol I	144.26	0.000	138.83	0.000	140.67	0.002	138.13	0.000	150.67	0.000	134.35	0.000	157.97	0.000
LDL-chol II	122.02		115.00		119.06		113.45		130.33		117.24		136.18	
Δ LDL-chol	↓ 22.23		↓ 23.83		↓ 21.61		↓ 24.68 ^(c)^		↓ 20.35 ^(c)^		↓ 17.12 ^(b)(b)^		↓ 21.79 ^(c)(c)(b)^	
HDL-chol I	40.19	0.000	42.26	0.000	37.78	0.000	43.98	0.000	37.75	0.000	35.71	0.008	38.66	0.000
HDL-chol II	50.33		53.22		51.66		53.83		46.91		45.46		47.55	
Δ HDL-chol	↑ 10.14		↑ 10.96		↑ 13.88		↑ 9.85 ^(c)^		↑ 9.16 ^(c)^		↑ 9.75 ^(b)(c)^		↑ 8.90 ^(b)(c)(c)^	
TGs I	147.95	0.000	154.29	0.006	138.56	0.563	160.32	0.006	140.45	0.023	134.24	0.077	143.24	0.109
TGs II	132.64		137.38		133.11		139.01		127.03		118.18		130.99	
Δ TGs	↓ 15.32		↓ 16.92		↓ 5.44		↓ 21.31 ^(c)^		↓ 13.42 ^(c)^		↓ 16.06 ^(b)(b)^		↓ 12.42 ^(b)(b)(c)^	

Abbreviations and measuring units: HbA1c—glycated hemoglobin (%); glycemia (mg/dL); MDRD-GFR—Modification of Diet in Renal Disease study equation for estimating glomerular filtration rate based on creatinine levels and patient characteristics (mL/min/1.73 m^2^); urea (mg/dL); creatinine (mg/dL); fibrinogen (mg/dL); CRP—C-reactive protein (mg/L); Total chol—total cholesterol (mg/dL); LDL-chol—low-density lipoprotein (mg/dL); HDL-chol—high-density lipoprotein (mg/dL); TGs—triglycerides (mg/dL). Symbols: Δ—difference from phase I to phase II; ↑ and ↓—increasing/decreasing levels. Superscript: ^(a)^ *p* < 0.001; ^(b)^ *p* < 0.05; ^(c)^ *p* > 0.05 for differences between the study (sub)groups.

## Data Availability

The data presented in this study are available on request from the corresponding author.

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
