# Peer review of "Post-Coronary Artery Bypass Grafting Outcomes of Patients with/without Type-2 Diabetes Mellitus and Chronic Kidney Disease Treated with SGLT2 Inhibitor Dapagliflozin: A Single-Center Experience Analysis"

_diagnostics, 2023, doi:10.3390/diagnostics14010016_

Round 1

Reviewer 1 Report

Comments and Suggestions for Authors

Thank the authors for their efforts. To improve the quality of the article, some aspects would have to be worked on.

1. Title: it is confusing when reading the paper, since it barely talks about SGLT2i beyond the Introduction. 

2. Parameters must be defined with their abbreviations the first time they appear. In this case: CABG (line 22), H-FABG (line 33). An initial section of abbreviations would make reading easier.

3. Indicate the real value of the p (line 40 and 208) as well as the magnitude of measurement of the GFRs (line 40).

4. Line 156: myocardial has to be written in Capital letter and, if previously described, remove the H-FABP description.

5. Section 2.3: The treatments used in the series of patients studied for each of the groups considered are not described or specified. This could influence the results described and should be reflected in the discussion.

6. Point 2.5: the statistical methodology used should be better described, both for discrete and continuous variables.

7. Line 178: specify if they were all patients, since in Table 1 it appears that there are patients with BMI less than 30.

8. The data presented in Table 2: I recommend making Figures, as they help understand the text.

9. Line 215: remove definition of the abbreviation if it has been done previously.

10. Review Table 2: There are letters (a,b,c) that are repeated in the data.

11. Line 242: specify the meaning of the comparison for each letter used, even if it appears already described in the text.

12. Discussion section: It is too long. Data is provided that corresponds to results (for example: paragraph 2, lines 346-349...). The data of the study are hardly discussed with the literature and that is why there is a lack of references. This section must be reviewed in depth and redrafted, specifying based on the objectives set out in the study.

13. The conclusion must be reviewed based on the previous sections. The SGLT2i appear again after the Introduction.

Author Response

Dear Reviewer,

Thank you very much for evaluating our manuscript. Your recommendations and comments have helped us improve our manuscript. Here we provide the requested corrections and address the comments. The changes we have made in the manuscript are highlighted in red.

1.Title: it is confusing when reading the paper, since it barely talks about SGLT2i beyond the Introduction.

SGLT2i was mentioned in material and methods line 134, results lines 242-245, discussion line 324,325, 349-359 and in conclusions line 401-405.

  1. Parameters must be defined with their abbreviations the first time they appear. In this case: CABG (line 22), H-FABG (line 33). An initial section of abbreviations would make reading easier.

            We mentioned the abbreviations in line 22 and 33, thank you for your comment.

  1. Indicate the real value of the p (line 40 and 208) as well as the magnitude of measurement of the GFRs (line 40).

Line 40; T2DM+CKD 7.01%-6.08% (p=0.000), T2DM-CKD 7.08%-6.04% (p=0.000), PreD+CKD 5.66%-4.98% (P=0.014) and PreD-CKD 6.03%-4.94% (p=0.00),

line 208 Mean ejection fraction increased by 8.43% overall it wents from 43.5% to 51.9% (p=0.000),

line 40 Compared to an overall gain of 11.51, the GFRs of patients with CKD improved by 18.93 (68.15%-87.07%, p=0.000) in the presence of established diabetes and 14.89 (64.75%-79.64%, p=0.000) in the prediabetes group.

  1. Line 156: myocardial has to be written in Capital letter and, if previously described, remove the H-FABP description.

Line 157 has been modified, thank you.

  1. Section 2.3: The treatments used in the series of patients studied for each of the groups considered are not described or specified. This could influence the results described and should be reflected in the discussion.

Dapagliflozin was administered in daily doses of 10 mg. Otherwise, the medication plans were individualized based on patients’ progress and other comorbidities (T2DM and related treatment). The treatment was standard for each group of patients with/without T2DM with/without CKD, we did not administer different classes of treatment so there will be no influence on final results in our study.

  1. Point 2.5: the statistical methodology used should be better described, both for discrete and continuous variables.

            The patient data were first anonymized and collected in Microsoft Office Excel version 2010, then processed in IBM SPSS Statistics for Windows, version 20 (IBM Corp., Armonk, NY, USA). Data series were organized in the following two main study groups and four subgroups based on the presence or absence of the two studied comorbidities:

  1. Patients with established type-2 diabetes mellitus (T2DM):
  2. With chronic kidney disease as well (T2DM+CKD);
  3. Without chronic kidney disease (T2DM-CKD).
  4. Patients with established or incidental prediabetes (PreD):
  5. With chronic kidney disease as well (PreD+CKD);
  6. Without chronic kidney disease (PreD-CKD).

The statistical analysis in the process of collecting and analyzing data were described as we mentiond in section 2.5.

  1. Line 178: specify if they were all patients, since in Table 1 it appears that there are patients with BMI less than 30.

            In terms of general cardiovascular risk factors, the patients’ overall body mass indices averaged just above the obesity threshold (30.12 kg/m2), we changed it in line 178.

  1. The data presented in Table 2: I recommend making Figures, as they help understand the text.

            Thank you for your comment, we mentioned echocardiographic parameters changing in fig.2, describing the significant data in table 2.

  1. Line 215: remove definition of the abbreviation if it has been done previously.

            The definition of the abbreviation was removed from line 215.

  1. Review Table 2: There are letters (a,b,c) that are repeated in the data.

            Uppercase: a) p<0.001; b) p<0.05; c) p>0.05 for differences between the study (sub)groups.

  1. Line 242: specify the meaning of the comparison for each letter used, even if it appears already described in the text.

We described the results in table 3 lines 242-286 and in lines 296-323.

  1. Discussion section: It is too long. Data is provided that corresponds to results (for example: paragraph 2, lines 346-349...). The data of the study are hardly discussed with the literature and that is why there is a lack of references. This section must be reviewed in depth and redrafted, specifying based on the objectives set out in the study.

            Thank you for all your indications, we redrafted the discussion adding references from literature that can improve our idea in this study.

  1. The conclusion must be reviewed based on the previous sections. The SGLT2i appear again after the Introduction.

            Thank you for your comment, we tried to write exactly the conclusion of our study after discussing the final results.

Reviewer 2 Report

Comments and Suggestions for Authors

How  was performed patient s  inclusion,  selection in study- was it randomized  study ?  If yes based on  which parameters ?   Or all patients  were included ?

 How patients  were included and  selected to study: was it randomized ? if yes  , based on  which data ?

If not  - are all patients passed CABG in mentioned period   included?

 It is not  correct to  access troponin decrease parameter  as  prognostic marker       from phase 1 to phase 2: it has tendency to  be  chronologically lower  at phase  2  after CABG related pathophysiologic   mechanisms, including  non cardiac  such as  renal , surgery, other... 

Please  open H-FABP   abbreviation in introduction tion initial    site.  Please indicate  in brief its  clinical significance in paper.

Comments on the Quality of English Language

 Can be improved  with more clear  scientific  statements 

Author Response

Dear Reviewer,

Thank you very much for evaluating our manuscript. Your recommendations and comments have helped us improve our manuscript. Here we provide the requested corrections and address the comments. The changes we have made in the manuscript are highlighted in red.

How was performed patient’s inclusion, selection in study- was it randomized study? If yes based on which parameters? Or all patients were included?

            Thank you for your comment, all patients were selected in our study, 120 patients, it is not randomized study, we included patients according to inclusion and exclusion criteria mentioned in the manuscript.

 How patients were included and selected to study: was it randomized? if yes, based on which data?

            Thank you for your comment, all patients were selected in our study, 120 patients, it is not randomized study, we included patients according to inclusion and exclusion criteria mentioned in the manuscript.

If not  - are all patients passed CABG in mentioned period included?

            Yes, all patients undergoing CABG who respected inclusion criteria were included in our study.

 It is not correct to access troponin decrease parameter as prognostic marker from phase 1 to phase 2: it has tendency to  be chronologically lower at phase 2 after CABG related pathophysiologic mechanisms, including non-cardiac such as renal, surgery, other...

Thank you for your important mention here; although in our study we wanted to mention the risk of reinfarction, that is why we measured cTnI in phase 2 among with H-FABP to study mention the relationship between these two cardiac biomarkers in reinfarction cases.

Please open H-FABP abbreviation in introduction initial site. Please indicate in brief its clinical significance in paper.

            We added the H-FABP abbreviation and its clinical significance in manuscript lines 98-101.

Round 2

Reviewer 1 Report

Comments and Suggestions for Authors

The authors have resolved the queries, improving the quality of the paper.

Author Response

Dear Reviewer,

Thank you again for reviewing our manuscript.

Reviewer 2 Report

Comments and Suggestions for Authors

  The  comments    should be  included in final version:

120 patients, it is not randomized study, 

All patients undergoing CABG who respected inclusion criteria were included in our study.

all patients undergoing CABG who respected inclusion criteria were included in our study.

Author Response

Dear Reviewer,

The suggested comments were included in the final version of the manuscript (line 325-328). Thank you again for reviewing our manuscript.